# Bayesian Optimized Meta-Learning for Uncertainty-Driven Optimization

## Abstract

This paper introduces a Bayesian-optimized meta-learning framework aimed at enhancing model performance in uncertain and noisy industrial environments. By integrating Bayesian Optimization with Model-Agnostic Meta-Learning (MAML), our approach dynamically fine-tunes model parameters for robust performance. The framework effectively identifies global minima despite uncertainty by utilizing Gaussian Process models with the Matérn kernel and the Maximum Probability of Improvement (MPI) acquisition function. Covariance analysis aligns training and validation losses, while L2 regularization prevents overfitting. Experimental results demonstrate the framework's ability to balance accuracy and generalization, making it suitable for diverse industrial optimization tasks.

## 1   Introduction

In many complex systems, particularly in industrial contexts like predictive maintenance, quality control, and process optimization, enhancing model performance under significant uncertainty and noisy data is a major challenge (1; 2; 3). For example, in predictive maintenance, sensors can generate noisy or incomplete data, complicating accurate equipment failure prediction (4; 5). These environments often require optimizing a function $f(\theta, D + \delta, \zeta)$, where $\theta \in \Theta \subset \mathbb{R}^p$ represents the control variables or model parameters, $D$ denotes the dataset, $\delta \in \mathbb{R}^d$ represents perturbations, and $\zeta$ accounts for noise. Both $\delta$ and $\zeta$ are treated as random variables with a specified joint probability density function $p(\delta, \zeta)$. The robust optimization problem can thus be formulated as:

$$\theta^{\star} = \arg\min_{\theta \in \Theta} \mathbb{E}_{\delta, \zeta} \left[ \mathcal{L}(\theta, D + \delta, \zeta) \right] + \lambda |\theta|_2^2, \tag{1}$$

subject to the constraints:

$$\Pr \left[ C_j(\theta, D + \delta, \zeta) \leq 0 \right] \geq 1 - \eta, \quad j = 1, \dots, d_c, \tag{2}$$

where $\mathcal{L}(\theta, D + \delta, \zeta)$ is the performance loss, and $\lambda |\theta|_2^2$ represents L2 regularization weighted by $\lambda$ to mitigate overfitting (6). The constraints $C_j(\theta, D + \delta, \zeta)$ must hold with a probability of at least $1 - \eta$, where $\eta \in (0, 1)$ represents the allowable risk of constraint violation or the probability of failure, ensuring robustness against data variations. Overfitting to noisy data is a significant risk, leading to poor generalization and unreliable performance. Traditional methods struggle to maintain both robustness and accuracy in such environments. This paper's key contribution is developing a Bayesian-optimized meta-learning framework to address these challenges. Bayesian Optimization (BO) effectively optimizes complex, noisy functions, particularly when gradient information is unavailable (7; 8; 9; 10). However, existing BO methods do not fully address the need for robustness under uncertainty, especially when combined with meta-learning techniques. We extend the meta-learning framework by integrating Bayesian Optimization (BO) with MAML (11). BO iteratively

optimizes meta-learning parameters using Gaussian Process (GP) models (12) to estimate loss functions under uncertainty. The GP models guide the optimization process toward the optimal parameter set. We introduce covariance analysis between training and validation losses to enhance robustness, measuring Bayesian risk and adjusting the model to reduce it. L2 regularization (6) is also incorporated to control overfitting, ensuring effective generalization across varying noise levels, making the framework a powerful tool for optimization in dynamic and uncertain environments.

## 2 Bayesian Optimized Meta-Learning

Our approach incorporates covariance analysis between training and validation losses, providing insights into the model's generalization capabilities. The methodology is outlined in Algorithm 1. In uncertain environments, the optimization process must account for perturbations and noise within the training dataset (9; 13). We define the input space as $x_e = (\theta, D + \delta, \zeta)$, where $\theta \in \Theta \subset \mathbb{R}^p$ represents the model parameters, $D$ denotes the dataset, $\delta \in \mathbb{R}^d$ represents perturbations, and $\zeta \in \mathbb{R}^n$ accounts for noise. These perturbations and noise are treated as random variables with a joint probability density function $p(\delta, \zeta)$. Gaussian Process (GP) models are used for their predictive and uncertainty estimation capabilities, essential for guiding optimization in uncertain settings (14). The objective function we seek to minimize is the expected loss function $J(\theta)$, expressed as:

$$J(\theta) = \int_\zeta \mathcal{L}(\theta, D + \delta, \zeta)p(\zeta)d\zeta, \tag{3}$$

where the integral represents the expectation over the noise space $\zeta$. The loss function $\mathcal{L}(\theta, D + \delta, \zeta)$ is modeled as a Gaussian Process (GP), which provides a probabilistic framework by capturing the relationship between different points in the parameter space through the covariance function (12). This covariance function plays a crucial role in understanding how changes in the loss function at one point impact the loss at another, particularly in the context of training and validation losses. Given a set of observations $y_1 = \{\mathcal{L}(\theta_i)\}_{i=1}^t$, the GP posterior distribution $J(\theta)$ is updated as:

$$J \sim GP(\mu_{\text{post}}^J(\theta), k_{\text{post}}^J(\theta, \eta)), \tag{4}$$

where $\mu_{\text{post}}^J(\theta)$ is the posterior mean and $k_{\text{post}}^J(\theta, \eta)$ is the posterior covariance function. These are derived from the prior GP using the observed data and are critical in guiding the optimization process. The acquisition function $\alpha_c(\theta)$ plays a crucial role in balancing exploration (searching in regions where the model is uncertain), and exploitation (focusing on regions where the model predicts low loss) (10). This balance is essential in Bayesian optimization, where the cost of evaluating the objective function is high. In this work, we utilize the Maximum Probability of Improvement (MPI) acquisition function, which prioritizes areas in the parameter space with the highest probability of improving upon the current best observation. The acquisition function can be defined as:

$$\alpha_c(\theta) = \alpha(\theta) \prod_{j=1}^{d_c} \Pr[C_j(\theta, D + \delta, \zeta) \leq 0], \tag{5}$$

where $\alpha(\theta)$ represents the probability of achieving an improvement over the best-known value (15). Although various acquisition functions such as Expected Improvement (EI), Entropy Search (ES), and Lower Confidence Bound (LCB) can be employed, MPI is particularly suited for scenarios where a conservative approach to optimization is required, focusing on areas with a high likelihood of yielding better results while effectively managing the exploration-exploitation tradeoff (16). To enhance generalization, we introduce covariance analysis between training loss $\mathcal{L}_{\text{train}}$ and validation loss $\mathcal{L}_{\text{val}}$. High covariance indicates strong generalization, as improvements in training loss translate to validation performance (17). The covariance is defined as:

$$\text{Cov}(\mathcal{L}_{\text{train}}, \mathcal{L}_{\text{val}}) = \mathbb{E}[(\mathcal{L}_{\text{train}} - \mathbb{E}[\mathcal{L}_{\text{train}}])(\mathcal{L}_{\text{val}} - \mathbb{E}[\mathcal{L}_{\text{val}}])], \tag{6}$$

A high covariance suggests that the model's performance on the training data reflects its performance on unseen data, indicating strong generalization. Conversely, low covariance may indicate overfitting or underfitting (18). This covariance analysis can be used to identify which components or feature extractors contribute most to generalization. For example, by decomposing the overall covariance

---

**Algorithm 1** Bayesian Optimization with L2 Regularization and Covariance Analysis

---

**Require:** Initial data $\mathcal{D}_{1:t} = \{x_{e1:t}, y_{1:t}\}$, GP prior $GP(\mu_{\text{prior}}^{\mathcal{L}}, k_{\text{prior}}^{\mathcal{L}})$, acquisition function $\alpha_c(\theta)$,
    regularization parameter $\lambda > 0$, iterations $T$
**Ensure:** Optimized parameters $\theta^\star$
  1: **Initialize** GP model with prior $GP(\mu_{\text{prior}}^{\mathcal{L}}, k_{\text{prior}}^{\mathcal{L}})$ using initial data $\mathcal{D}_{1:t}$
  2: **for** $t = 1$ to $T$ **do**
  3:     Update GP posterior $J(\theta) \sim GP(\mu_{\text{post}}^{J}(\theta), k_{\text{post}}^{J}(\theta, \eta))$
  4:     Compute regularized expected loss $J(\theta) = \int_\zeta \left[ \mathcal{L}(\theta, D + \delta, \zeta) + \lambda|\theta|_2^2 \right] p(\zeta)d\zeta$
  5:     Recompute $\text{Cov}(\mathcal{L}_{\text{train}}, \mathcal{L}_{\text{val}})$
  6:     Optimize acquisition function $\theta_{t+1} = \arg\max_{\theta \in \Theta} \left[ \alpha(\theta) + \beta\text{Cov}(\mathcal{L}_{\text{train}}, \mathcal{L}_{\text{val}}) \right]$
  7:     Obtain new observation $y_{t+1} = \mathcal{L}(\theta_{t+1}) + \epsilon_{t+1}$
  8:     Update dataset $\mathcal{D}_{1:t+1} \leftarrow \mathcal{D}_{1:t} \cup \{x_{et+1}, y_{t+1}\}$
  9: **end for**
10: **return** Optimized parameters $\theta^\star$

---

into contributions from different components of the model:

$$\text{Cov}(\mathcal{L}_{\text{train}}, \mathcal{L}_{\text{val}}) = \sum_{i=1}^{m} \text{Cov}(\mathcal{L}_{\text{train}}^{(i)}, \mathcal{L}_{\text{val}}^{(i)}), \tag{7}$$

where $\mathcal{L}_{\text{train}}^{(i)}$ and $\mathcal{L}_{\text{val}}^{(i)}$ represent the losses associated with the $i$-th component and are used to modify the acquisition function, prioritizing parameter regions that ensure generalization:

$$\alpha_c(\theta) = \alpha(\theta) + \beta\text{Cov}(\mathcal{L}_{\text{train}}, \mathcal{L}_{\text{val}}), \tag{8}$$

where $\beta$ is a weight that determines the influence of the covariance on the acquisition function. This modification ensures that the search process not only seeks to minimize the loss but also to find solutions that generalize well to unseen data. L2 regularization penalizes large parameter values, preventing overfitting and enhancing robustness (19; 20). The regularized objective function is:

$$\theta^\star = \arg\min_{\theta \in \Theta} \mathbb{E}_{\delta, \zeta} \left[ \mathcal{L}(\theta, D + \delta, \zeta) + \lambda|\theta|_2^2 \right], \tag{9}$$

where $\lambda|\theta|_2^2$ controls model complexity, promoting stability in noisy environments. This regularization complements our covariance-based approach by promoting consistency between training and validation performance, which is critical for generalization. The regularized loss function is updated within the GP model as:

$$\mathcal{L}(\theta) = \mathcal{L}_{\text{observed}}(\theta) + \lambda|\theta|_2^2, \tag{10}$$

with iterative updates of the GP model, acquisition function, and subsequent selection of query points $\theta_{t+1}$. This process continues, refining the search for optimal parameters $\theta^\star$ while ensuring robustness and generalization. This methodology systematically refines parameter search through GP models, L2 regularization, and covariance analysis, ensuring that the optimized parameters $\theta^\star$ are robust to uncertainties and generalize well to unseen data.

## 3   Empirical Analysis

We employed the Matérn kernel within a Gaussian Process (GP) framework to minimize Bayes risk under noisy conditions. The Matérn kernel is a flexible kernel function commonly used in Gaussian Processes, with the smoothness parameter $\nu$ controlling the level of smoothness of the function it models. By adjusting $\nu$, the Matérn kernel can model functions ranging from very rough (low $\nu$) to very smooth (high $\nu$), making it well-suited for environments where uncertainty impacts outcomes. Combined with the Maximum Probability of Improvement (MPI) acquisition function, our approach successfully navigates the optimization landscape, converging on global minima even with varying noise levels. We validated the Matérn kernel's robustness by comparing it to the Radial Basis Function (RBF) kernel using mean squared error (MSE) and log-likelihood metrics. The Matérn

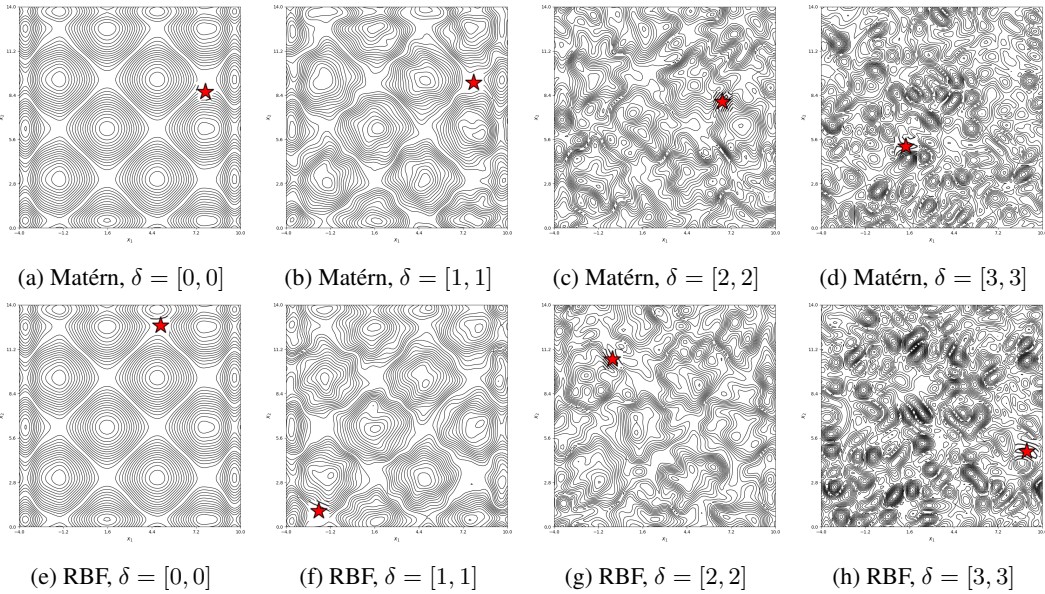

| (a) Matérn, $\delta = [0, 0]$ | (b) Matérn, $\delta = [1, 1]$ | (c) Matérn, $\delta = [2, 2]$ | (d) Matérn, $\delta = [3, 3]$ |

| (e) RBF, $\delta = [0, 0]$ | (f) RBF, $\delta = [1, 1]$ | (g) RBF, $\delta = [2, 2]$ | (h) RBF, $\delta = [3, 3]$ |

Figure 1: Comparison of global minima identification using Matérn and RBF kernels under varying noise conditions ($\delta$). The red star represents the global optima. The Matérn kernel shows more accurate and consistent minima detection across all noise levels compared to the RBF kernel.

kernel achieved a lower MSE (19.84 vs. 29.34 for RBF) and higher log-likelihood values across noise levels, demonstrating its superior handling of noisy data. ts tunable $\nu$ enables effective management of short and long range dependencies, ensuring focus on patterns rather than noise induced anomalies. Our contour plots, shown in Figure 1, illustrate the optimization function's evolution as the noise perturbation parameter $\delta$ varies. These plots were generated using real data, consisting of 50% good images and 50% anomalous images. As $\delta$ increases, the function's topology transitions from multiple local minima to fewer, distinct global minima, highlighting the Matérn kernel's effectiveness in managing increased data point distances and reducing noise influence. Covariance analysis between training and validation losses further guides the optimization process, favoring regions in the parameter space that yield reliable global minima despite increased noise. The Matérn kernel's ability to adjust dynamically to varying noise levels ensures the GP model remains attuned to true underlying patterns, making it highly suitable for applications requiring robust decision-making under uncertainty, such as in real-time industrial systems. This methodology, integrating the Matérn kernel within a GP framework and leveraging MPI, significantly advances the robust optimization of models in noisy, uncertain environments, with potential applications across various industrial settings.

## 4    Conclusion and Future Work

We proposed a Bayesian-optimized meta-learning framework tailored for improving model performance in noisy and uncertain industrial environments. By integrating Gaussian Processes (GP) with the Matérn kernel, our approach demonstrated strong robustness and accuracy in identifying global minima across varying noise levels. The use of the Maximum Probability of Improvement (MPI) acquisition function and covariance analysis facilitated effective optimization in complex landscapes. Our results highlight the framework's capacity to maintain a balance between accuracy and generalization, which is essential for reliable performance in real-world applications. The inclusion of L2 regularization further enhanced the model's ability to avoid overfitting in dynamic conditions. However, the approach may face scalability challenges when applied to extremely large datasets or real-time systems due to the computational demands of Bayesian Optimization and Gaussian Processes. This work contributes to the field of robust optimization under uncertainty, with potential applications in predictive maintenance, quality control, and process optimization. Future research may explore other meta-learning algorithms or alternative kernels to further enhance the adaptability and effectiveness of the framework in even more challenging scenarios.

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

# A    Appendix / supplemental material

Optionally include supplemental material (complete proofs, additional experiments and plots) in appendix. All such materials **SHOULD be included in the main submission.**

