# OpenReview forum: "Bayesian Optimized Meta-Learning for Uncertainty-Driven Optimization"
_NeurIPS.cc/2024/Workshop/BDU — Submitted to NeurIPS BDU Workshop 2024_

### Official Review · Reviewer_ueia · 2024-09-19
**Review of the article**

**Rating:** 2
**Confidence:** 4

**Review:**

I was quite excited by the abstract and what the article would offer to the workshop. Unfortunately I then read the article. It is not ready. I think there are interesting aspects to the work, but the article is not sufficiently well written for it to make much sense. My concerns are as follows:

- The abstract claims we will combine MAML with BO. But at no point in the article is it explained what is meant by MAML.
- In the introduction there is a dataset D, but it is not explained what this is, or even the perturbations or noise for that matter. The problem is not well specified. It's a problem because in GP/BO world, including in Algorithm 1, we have data D_{1:t}. I can't work out whether these are the same datasets or not.
- In the paragraph starting at l48, the first line has a loss function L(theta, D+delta,zeta). The last line has loss functions depending only on theta. This might look like a minor thing, but the fact that D, delta and zeta all vanish whenever BO is considered make it impossible to work out what is going on.
- in (5), I can't work out what this probability is over. All of D, delta and zeta are present here; I suspect the latter two are random but it's far from clear. How the probabilities are estimated is really important and entirely unexplained. The product also indicates an implicit assumption that the constraints are satisfied or not independently of each other, which seems unlikely.
- In (6), where do L_train and L_val come from? So far there has been no description of how the models are trained or fit.
- In Alg 1, we once again have L parameterised by theta, D, delta and zeta, then by just theta.
- In (8) the loss covariance is not dependent on theta, so can have no impact on the theta that will be selected. Since this makes no sense, I can only assume that there is theta-dependence. But then the question arises for how does that covariance, as a function of theta, get estimated?
- In (10) we suddenly get yet more regularisation for some reason. Regularisation is good. But I am unconvinced it has any place as a "feature" in a BO routine. It would be better included in the loss functions.
- Section 3 is not particularly interesting or novel for a BO workshop. The space should have been used to better explain the core problem.

I am sorry not to be more positive. As I said at the start, I think the topic is interesting, but you really need to explain things more completely and accurately for it to make sense.

---

### Official Review · Reviewer_g2Bn · 2024-09-27

**Rating:** 5
**Confidence:** 3

**Review:**

The idea of integrating BO with MAML is interesting. I hope this feedback will be useful to the authors as they continue to develop this idea. I recognize that much of what I describe as being unclear is probably due to the space limitation, but there are some parts that are unclear to me that are pretty clearly important.

* The connection to MAML is central to the paper, yet not very clear. I assume this is happening in the fine tuning stage?

* The notation is unclear. L is sometimes described as a function of just theta, other times (theta, D+\delta, \zeta).

* The use of Probability of Improvement is surprising, it is not a very common choice in BO. Readers who work in BO will want to know more about this choice, and to see a comparison with EI and UCB.

* It sounds like the kernel is not ARD as there is description of a single parameter for determining smoothness? That is another choice that a reader familiar with BO would need to see justified.

* The description of (10) makes it sound like the GP is being fit to the regularized loss function, but then it wasn't clear later. I'd expect the GP to be fit to the unregularized loss function.

---

### Decision · Program_Chairs · 2024-10-09

**Decision:**

Reject

**Comment:**

As the reviewers pointed out, it is not clear how MAML relates to the proposed method, despite that "meta-learning" was significantly emphasized in this paper. While we encourage submitting ongoing work to the workshop, this paper requires a lot more effort to justify the intended contribution and novelty. That said, we do think the abstract and idea are very interesting. We suggest the authors take into account the reviews and continue pursuing this direction.